# A New Approach to Automatically Calibrate and Detect Building Cracks

**Zongchao Liu [1], Xiaoda Li [1], Junhui Li [1,\*] and Shuai Teng [2,\*]**

[1] School of Civil Engineering, Guangzhou University, Guangzhou 510006, China; zhjtlzc@gzhu.edu.cn (Z.L.); xiaodali@gzhu.edu.cn (X.L.)

[2] School of Civil and Transportation Engineering, Guangdong University of Technology, Guangzhou 510006, China

[\*] Correspondence: lijunhui@gzhu.edu.cn (J.L.); 1112009002@mail2.gdut.edu.cn (S.T.); Tel.: +86-13724053357 (J.L.); +86-15914485576 (S.T.)

**Abstract:** Timely crack detection plays an important role in building damage assessment. In this study, an automatic crack detection method based on image registration and pixel-level segmentation (improved DeepLab_v3+) is proposed. Firstly, the moving images are calibrated by image registration, and the similarity method is adopted to evaluate the calibrated results. Secondly, the DeepLab_v3+ is improved and used to segment the fixed images and the calibrated images. Finally, the difference of crack pixels between the fixed and calibrated images is estimated, and the key parameter is investigated to find the optimal optimizer and learning rate. The results illustrate that: (1) the image registration technology shows excellent calibration achievement and the average error is only 4%; (2) with the resnet50 being selected as the backbone network of improved Deeplab_v3+, the automatic detection method proposed in this study is more efficient in comparison with other common pixel-level segmentation algorithms; (3) the best network optimizer of improved Deeplab_v3+ and learning rate of crack segmentation task are sgdm and 0.001, respectively. The crack detection method proposed in this study can significantly improves the technical level of crack detection in practical projects.

**Keywords:** structural health monitoring; crack detection; image registration; improved DeepLab_v3+; pixel-level segmentation

## 1. Introduction

External cracks of aging infrastructures (such as buildings (Figure 1), bridges, and pavements, etc.) are potential dangers for the structural durability and safety. Cracks of different degrees commonly appear in building structures. The structural cracks are mainly caused by an inadequate bearing capacity of the structures. It is the characteristic of the structural damage initiation or the symptom of insufficient structural strength, which is relatively dangerous, and the cracks must be further analysed to avoid the following disasters. The non-structural cracks, including temperature cracks and shrinkage cracks, always have little impact on the bearing capacity of the structure. Although these non-structural cracks do not reach the dangerous level of building collapse, these non-structural cracks can cause leakage, corrosion, concrete carbonation, etc., resulting in the reduction of the durability of building components, and even a serious potential threat to the safety and reliability of the structure. Structural health monitoring (SHM) is an effective way to recognize the cracks and evaluate the degree of damage, and structural maintenance can be subsequently proposed to prevent further crack propagation. However, the traditional manual visual inspection of cracks is labor-intensive, subjective, and error-prone, which can hardly meet the long-term development requirements for the detection of large-scale and complex modern structures. Therefore, the automatic and efficient crack detection method becomes an urgent need.

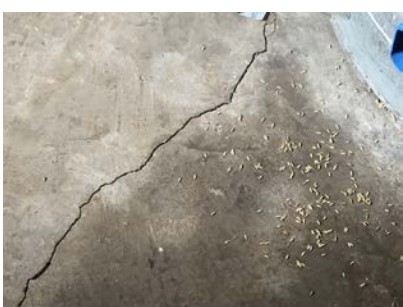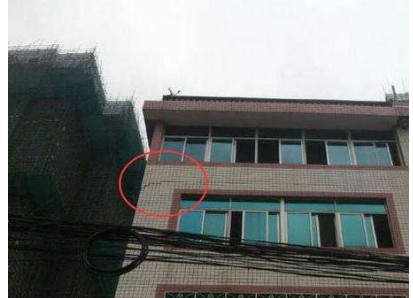

**Figure 1.** Building crack cases.

Recently, artificial intelligence (AI) algorithms are developing rapidly and provide great convenience for the automatic crack detection. For instance, the artificial neural network (ANN) technology has been explored in detecting the rail surface cracks and potholes of asphalt pavement surfaces [1,2]. However, many disadvantages, such as a slow convergence, over-fitting, and a high computational cost are also exposed in practical applications [3,4]. Therefore, a fast and high-precision detection technology is still urgently needed.

Deep learning (DL) provides a more advanced method for SHM with a high computational performance and accuracy. As a representation of DL algorithms, the deep convolutional neural network (DCNN) has been widely used in SHM. It can extract features from the original data automatically and obtain advanced features through multiple processing layers, gradually [5]. Meanwhile, the DCNN has a faster computing speed and a better robustness for the usage of the partial connection of neurons and pooling operations (down-sampling), which leads it to be an effective SHM method. The application of DCNN in image classification for pavement cracks, sewer defects, and road damage detection exhibits excellent performances [6–9]. Furthermore, the sliding window method is also employed to obtain the crack location [10,11]. However, this method always results in high computational costs as every window needs to be classified. Object detection technology can obtain the crack locations more accurately through creating a bounding box around the interest region. In the field of SHM, two common algorithms are used for objects detection: (1) a two-stage model, i.e., region-based CNN series, RCNN, Fast-RCNN, and Faster-RCNN, which have been used to detect the post-event building [12], concrete structure [13], and asphalt pavements [14]; and (2) a one-stage model, i.e., the you only look once (YOLO) and single shot multi-box detector (SSD) applied to detect the cracks (both in bridges and pavements) [15,16] and road defects [17], which show faster processing speeds than that of the two-stage model [9].

The pixel-level segmentation algorithm has been widely concerned for it can further improve the precision of defect information. It identifies the pixel distribution of the object, which can be used to analyze the object features (e.g., crack length, width, and area). Several kinds of neural networks have been developed to automatically implement pixel-level crack detection [18–22]. Compared with image classification and object detection, pixel-level segmentation is more effective and accurate in providing information about the distribution path and the shape of cracks. These advantages provide the potential for deeply extracting pixel-level quantifiable information of crack features. The latest version of the DeepLab_v3+ combines the advantages of the spatial pyramid pooling (SPP) and encoder–decoder structure [23]. It provides the excellent pixel-level segmentation method and has been successfully used for human and animal, skin and smoke detection [23–25]. In the field of SHM, the DeepLabv3+ has also been used to detect road potholes [26] and cracks [27] automatically, and its high detection accuracy has been verified. However, there is still a huge challenge in using intelligent algorithms for the automatic detection of crack images and identify and merge the different crack images captured at different views. It is difficult to ensure whether the camera is located at the same position [28,29]. Therefore, it is necessary to calibrate images in order to accurately obtain the crack change information of

images from different perspectives. Image registration is an image processing technology that aligns two or more images of the same scene with respect to a particular reference image (fixed image), and it has been widely used in remote sensing [30], medicine [31–33], and other fields for its high precision.

Therefore, this study proposes an automatic approach to detect the change of cracks in the consideration of view-influence based on the image registration and pixel-level segmentation technology (DeepLab_v3+). Firstly, the moving images are calibrated by image registration, and the similarity method is adopted to evaluate the calibrated results. Secondly, the DeepLab_v3+ will be improved and subsequently used to segment the fixed images and the calibrated images. Finally, the difference of crack pixels between the fixed and calibrated images is finally estimated, and the key parameter is investigated to find the optimal optimizer and learning rate. This study can help to improve the efficiency of crack detection in practical projects.

## 2. Materials and Methods

The whole process, including image registration and crack detection, was conducted in MATLAB (MathWorks Inc., Natick, MA, USA). A total of 1100 moving images were obtained through translating, rotating, and scaling 100 fixed images. The improved Deeplab_v3+ was obtained with the comparison of different backbone networks and then used to detect the change of cracks. The optimal optimizer and learning rate of the detection method in this study were obtained through parameter analysis.

### 2.1. Image Registration

Image registration is an image processing technique that can align the fixed and moving images. The critical work of image registration is to solve the transformation matrix (**T**), which describes the transformation information between the fixed and moving images. Generally, the transformation is defined in the following:

$$\begin{bmatrix} x' \\ y' \\ 1 \end{bmatrix} = \begin{bmatrix} a_1 & a_2 & a_5 \\ a_3 & a_4 & a_6 \\ 0 & 0 & 1 \end{bmatrix} \begin{bmatrix} x \\ y \\ 1 \end{bmatrix} \tag{1}$$

where, $\begin{bmatrix} a_1 & a_2 & a_5 \\ a_3 & a_4 & a_6 \\ 0 & 0 & 1 \end{bmatrix}$ is the transformation matrix (**T**), $(x, y)$ and $(x', y')$ are the locations before and after the transformation, respectively. The rotational matrix including $a_1$, $a_2$, $a_3$, and $a_4$ will lead to the rotation of the image, and $a_1$ and $a_4$ also represent the magnification (reduction) of the $x$ and $y$ coordinates of the image, respectively; $a_5$ and $a_6$ represent the translation of the $x$ and $y$ coordinates of the image, respectively. Therefore, it is critical to determine the six parameters accurately in order to implement image registration. A two-step model was established in this study:

**Step 1:** evaluation of image similarity. Mutual Information (*MI*) is used to assess the similarity of image intensities measured between the fixed and moving images. The maximized *MI* means the moving images have been accurately aligned.

$$MI(x, y) = \sum_{y \in Y} \sum_{x \in X} p(x, y) \log\left(\frac{p(x, y)}{p_1(x) p_2(y)}\right) \tag{2}$$

where, $p(x, y)$ is the joint distribution function and $p_1(x)$, $p_2(y)$ is the marginal distribution functions of the two random variables $(X, Y)$, $MI \subseteq [0,1]$. In this study, $X$ and $Y$ are fixed and moving images, respectively. If $MI = 1$, the two images coincide completely while they are irrelevant with $MI = 0$.

**Step 2:** optimization of key parameters. The optimization algorithm based on gradient descent function is employed to obtain accurate image transformation parameters. The

gradient descent adopts downhill steps proportional to the local gradient of the cost function *MI*:

$$a_i^{j+1} = a_i^j - k\frac{\partial MI}{\partial a_i}\Big|a_i^j \tag{3}$$

where, $i = 1, 2 \dots, 6$ (i.e., six transformation parameters), $j$ is the number of iterations, and $k$ is the relaxation factor ($0 < k < 1$, in this study, $k = 0.5$).

The image registration process is: (1) the initial similarity of the fixed and moving images is obtained through the mutual information (Equation (2)). (2) The initial transformation matrix is updated by the gradient descent algorithm (Equation (3)), and the new transformation matrix is used for image transformation to obtain a new image. (3) The similarity between the new image and fixed image is evaluated again. The steps (2) and (3) are repeated until the maximum iterations are obtained. The above process has been shown in Figure 2.

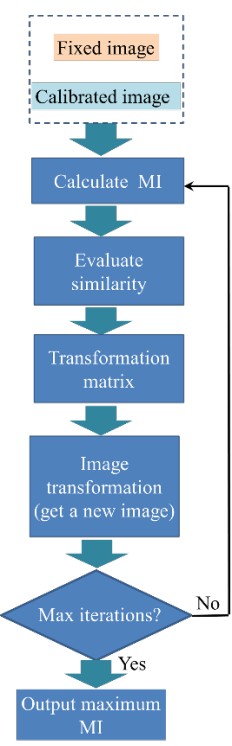

**Figure 2.** Image registration process.

### 2.2. DeepLab_v3+

The DeepLab_v3+ is a network model containing an encoder–decoder structure (Figure 3), which implements the pixel-level segmentation task. It is evolved based on DeepLab_v3 [34]. The DeepLabv3+ network employs a DCNN (backbone network) as a feature extractor to extract the feature information of objects. The object prediction results are subsequently obtained through the process of the specific encoder–decoder structure.

In the encoder module, the atrous spatial pyramid pooling (ASPP) sub-module, including four atrous convolution layers and one pooling layer, is used to extract features and reduce data dimensions. Finally, a $1 \times 1$ convolution kernel is used to extract features from the above information (ASPP sub-module), which is used as a branch input of the decoder. The atrous convolution is used to capture multi-scale information by obtaining the filter's field-of-view using different convolution kernel sizes, which is a generalization of the standard convolution. It is defined as follows:

In the case of two-dimensional raw data (Figure 4), for each location $i$ on the output feature map $y$ and a convolution filter $w$, the atrous convolution is applied over the input feature map $x$ as follows:

$$y[i] = \sum_k x[i + r \cdot k]w[k] \tag{4}$$

where, the atrous rate $r$ determines the stride with which we sample the input data ($r = 2$ in Figure 4). Note that standard convolution is a special case where $r$ is 1. The filter's field-of-view is adaptively modified by changing the rate value.

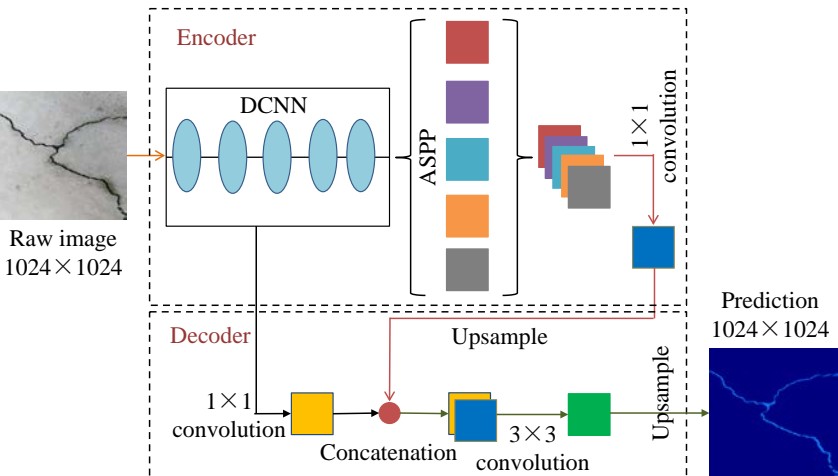

**Figure 3.** The DeepLab_v3+ network.

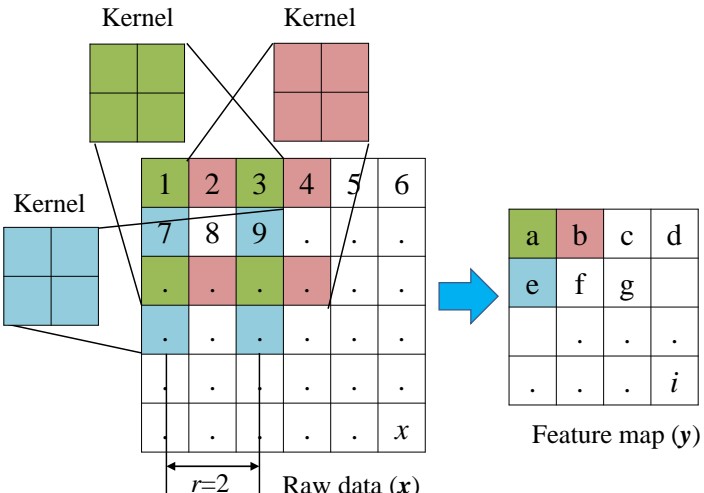

**Figure 4.** Atrous convolution process.

In the decoder module, the transposed convolution is used to extend the dimension of the feature map. Another branch input of the decoder comes from the backbone network and the special convolution operation ($1 \times 1$ convolution kernel). After the concatenation, convolution ($3 \times 3$ convolution kernel), and up-sampling, the feature maps are gradually restored to their original spatial dimensions, the output layer outputs each pixel classification of the raw image. The cracks have been marked and the pixel-level segmentation of the object region has been finished now.

### 2.3. Crack Change Detection

The detection of crack change before and after image registration is mainly divided into three steps: (1) align the moving images refer to the fixed images; (2) segment crack

pixels by the improved DeepLab_v3+; (3) calculate crack change ratio of the fixed and moving images.

**(1) Calibration of the moving images**

In this section, 100 crack images (1024 × 1024 pixels) were used as fixed images. Firstly, these images were scaled (Dataset A in Table 1), translated (Dataset B), rotated (Dataset C), and hybrid transformed (Dataset D and Dataset E). The images after transformation were named as moving images (total 1100 images). Table 1 shows the classification of the datasets and their detailed transformation paths. The moving images aligned with the fixed images were named as calibrated images.

**Table 1.** Moving image library.

| Dataset | Transformation Method | Motion Parameters | | | Sample Number |
|---------|----------------------|-------------------|---|---|---------------|
| A | Scaling | Factor = 0.8 | Factor = 0.9 | Factor = 1.1 | 300 |
| B | Translation (pixel) | [x = −10, y = −10] | [x = −30, y = −30] | [x = −50, y = −50] | 300 |
| C | Rotation | −5 degrees | −15 degrees | −25 degrees | 300 |
| D | Translation (pixel) and Rotation | [x = −10, y = −10] and −10 degrees | | | 100 |
| E | Scaling and Translation (pixel) and Rotation | 1.1 and [x = −10, y = −10] and −10 degrees | | | 100 |
| Total | | | | | 1100 |

Note: "−" is shift left.

**(2) Crack segmentation by the DeepLab_v3+**

As a deep learning model, the DeepLab_v3+ needs network training prior to the designative detection task. In this study, 1200 crack images (1024 × 1024 pixels) were collected, and the "Image Labeler" toolbox was used to label these cracks (Figure 5). Among these images, 75% of them were used as training images, and the rest were used as testing images to evaluate the effect of crack segmentation.

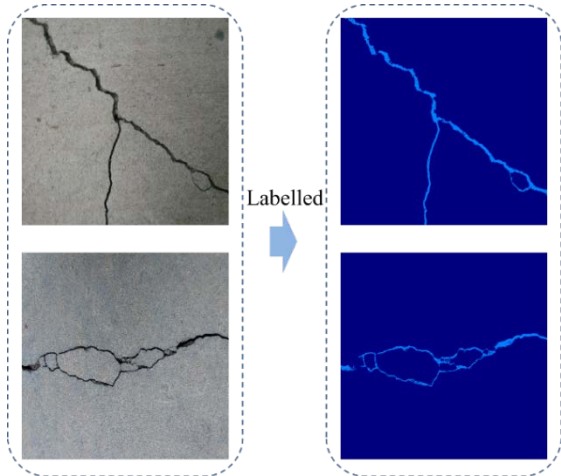

**Figure 5.** Labelled cracks.

The DeepLab_v3+ needs a backbone network (the DCNN in Figure 3) to extract the features of the object, and there are several backbone networks can be combined with the DeepLab_v3+ besides the official 'xception' backbone network. However, the effect of different backbone networks on exclusive crack detection is not clear. Therefore, the influence of backbone networks on the results should be firstly investigated. This study employed five well-known DCNNs ('resnet18', 'resnet50', 'mobilenetv2', 'xception', 'inception-resnetv2') as the backbone network of the DeepLab_v3+ for crack segmentation. The testing images were used to evaluate the crack segmentation results and identify the optimal DCNN model. Subsequently, the optimal model (improved DeepLab_v3+) was

used to implement the crack segmentation (300 testing images). In order to prove the feasibility of the improved DeepLab_v3+, this study will compare the results with these obtained from popular SegNet, FCN, and U-Net network models.

Intersection over Union (IoU) was an evaluation indicator of the DeepLab_v3+ for evaluating the overlap between the predicted ($A_p$) and real ($A_r$) object pixels. IoU was defined as:

$$IoU = \frac{\text{area}(A_p \cap A_r)}{\text{area}(A_p \cup A_r)} \tag{5}$$

and the Mean IoU of all classes, defined as:

$$MIoU = \frac{IoU}{N} \tag{6}$$

where, $N$ is the number of classes. In this study, $N = 2$, which means crack and non-crack classes were contained.

The *accuracy* and *F−score* were used to evaluate the classification effect:

$$Accuracy = \frac{TP + TN}{TP + FP + FN + TN} \tag{7}$$

$$Precision = \frac{TP}{TP + FP} \tag{8}$$

$$Recall = \frac{TP}{TP + FN} \tag{9}$$

$$F - score = 2 \times \frac{Precision + Recall}{Precision \times Recall} \tag{10}$$

where, true positive (*TP*): a real crack pixel is predicted correctly. False positive (*FP*): s real non-crack pixel is predicted as a crack pixel. False negative (*FN*): a real crack pixel is predicted as a non-crack pixel. True negative (*TN*): a real non-crack pixel is predicted correctly.

**(3) Calculation of crack change rate**

Assuming that the crack pixel number in image *i* is *N*, and in the entire image *i* is *I*, the ratio of crack pixels in the entire image is defined as:

$$Ratio = \frac{N}{I} \times 100\% \tag{11}$$

Therefore, the crack change ratio of the fixed image *i* and its corresponding moving image is:

$$Dif\_Ratio = Ratio_{Fixed} - Ratio_{Moving} \tag{12}$$

where, $Ratio_{Fixed}$ and $Ratio_{Moving}$ are the ratios of crack pixels in the fixed and moving images, respectively.

## 3. Results and Discussions

### 3.1. Image Registration Results

A total of 1100 moving images (sample library in Section 2.3) were aligned by the image registration technology, and their similarities related to the fixed mages were calculated. A part of the calibration results is shown in Figure 6. Detailed similarities of scaling, translation, and rotation images are exhibited in Figures 7–9, respectively. Figure 10a,b show the registration similarity of the hybrid transform images (translation and rotation for Figure 10a and scaling, translation, and rotation, for Figure 10b). Due to the movement of the image, the black area in the calibrated image is lost. It can be summarized from the results that the fixed and calibrated images have a high similarity (the average similarity is higher than 0.8).

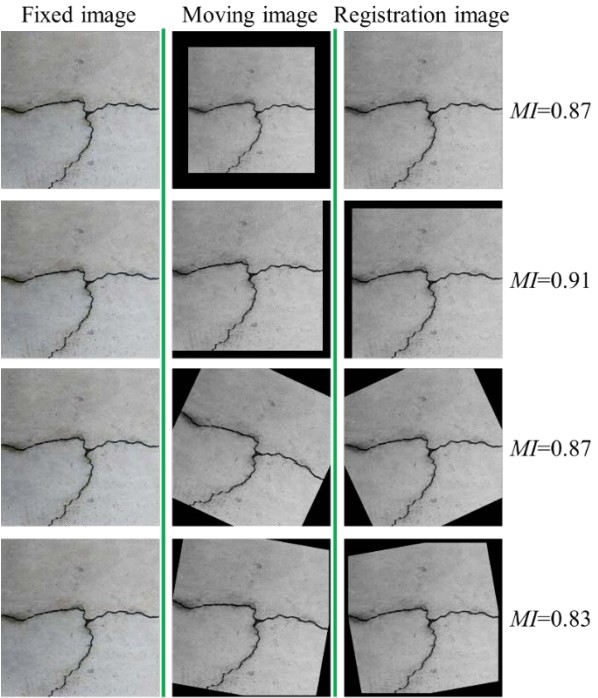

**Figure 6.** Image registration results of the cracks.

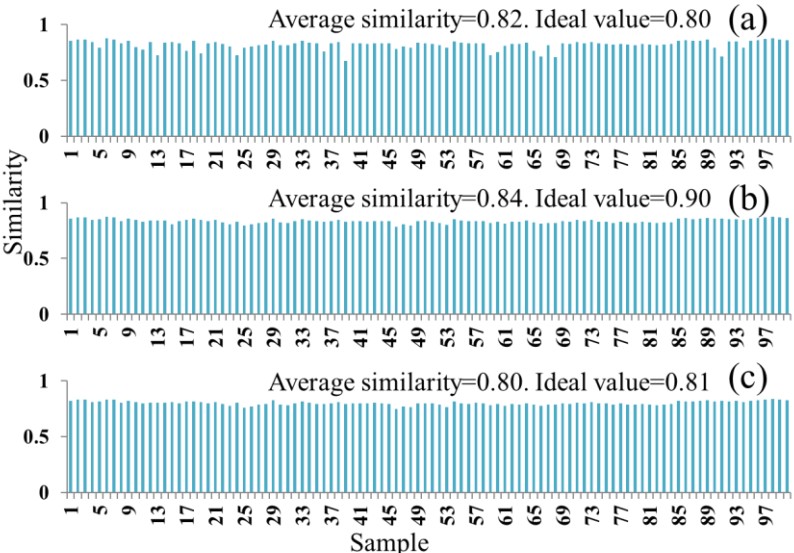

**Figure 7.** The registration similarity of the scaling images. (**a**) Scaling factor = 0.8; (**b**) scaling factor = 0.9; and (**c**) scaling factor = 1.1.

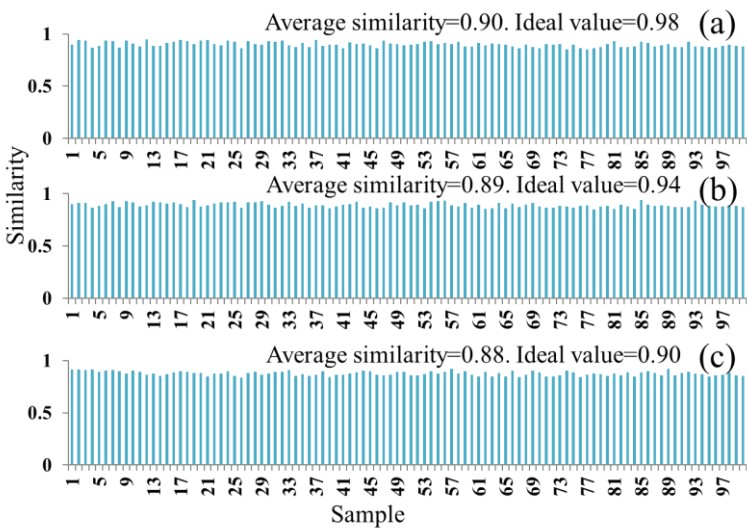

**Figure 8.** The registration similarity of the translation images. (**a**) [x = −10, y = −10]; (**b**) [x = −30, y = −30]; and (**c**) [x = −50, y = −50].

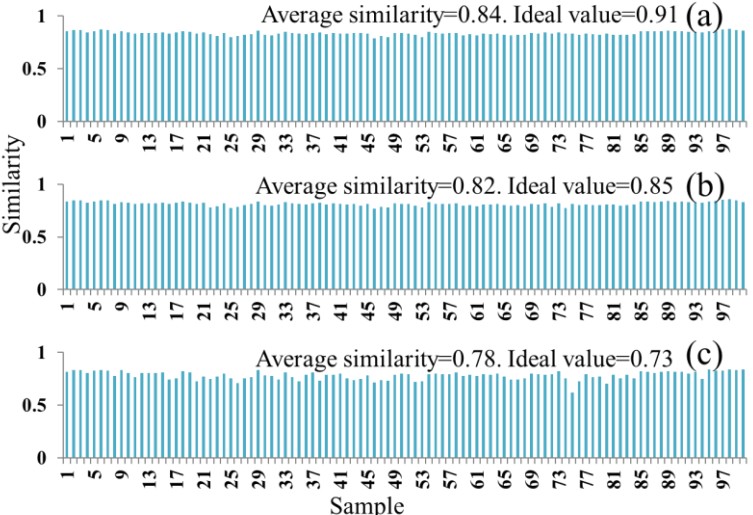

**Figure 9.** The registration similarity of the rotation images. (**a**) Rotation angle = −5; (**b**) rotation angle = −15; and (**c**) rotation angle = −25.

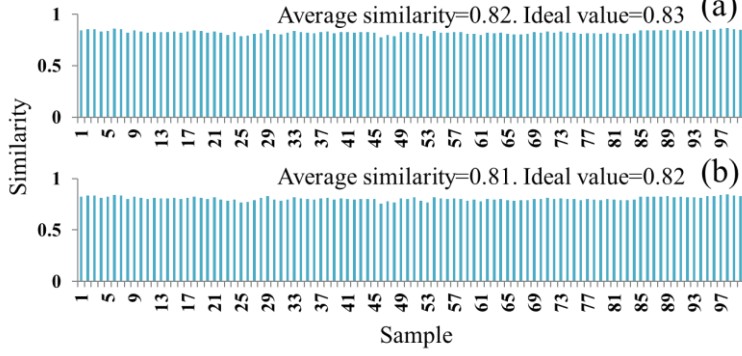

**Figure 10.** The registration similarity of the hybrid transformed images. (**a**) Translation and rotation; (**b**) scaling, translation, and rotation.

Table 2 shows the relative error of image registration. The relative errors of all datasets are less than 10%, and the average error is 4%. In addition, the image registration shows the

excellent registration results for the hybrid transform images with the relative error being only 1%. Generally, the similarity of the calibrated image can reach more than 80% accuracy despite the image having a single movement or a complex multi-directional coupling movement. The results in Table 2 prove that the image registration method can be used to accurately align the moving images with the fixed images.

**Table 2.** Relative error of image registration.

| Dataset | Transformation Method | Relative Error | | |
|---------|----------------------|------|------|------|
| A | Scaling | 3% | 7% | 1% |
| B | Translation | 8% | 5% | 2% |
| C | Rotation | 8% | 4% | 4% |
| D | Translation and Rotation | 1% | | |
| E | Scaling and Translation and rotation | 1% | | |

### 3.2. Crack Detection Results

The detection results of five different DeepLab_v3+ networks (('resnet18', 'resnet50', 'mobilenetv2', 'xception', and 'inception-resnetv2') used as the backbone network, respectively) are listed in Table 3. The 'resnet50' achieves the highest MIoU, accuracy, and F-score simultaneously; some detection examples are shown in Figure 11. Interestingly, the 'mobilenetv2' has the fastest computing speed at the expense of a small amount of precision. In the most extreme model, the 'inception-resnetv2' achieves the lowest accuracy and consumes a lot. Therefore, the DeepLab_v3+ adopting 'resnet50' as the backbone network was selected as the crack detection approach (named as: improved DeepLab_v3+).

**Table 3.** Segmentation results of different DeepLab_v3+ models.

| Evaluation Indicators | Backbone Network | | | | |
|-----------------------|----------|----------|------------|---------|-----------------|
| | **Resnet18** | **Resnet50** | **Mobilenetv2** | **Xception** | **Inceptionresnetv2** |
| MIoU | 0.82 | 0.84 | 0.82 | 0.80 | 0.75 |
| Accuracy | 99% | 99% | 99% | 99% | 99% |
| F-score | 0.91 | 0.93 | 0.90 | 0.88 | 0.81 |
| Detection time | 1158 s | 1783 s | 649 s | 1397 s | 2825 s |

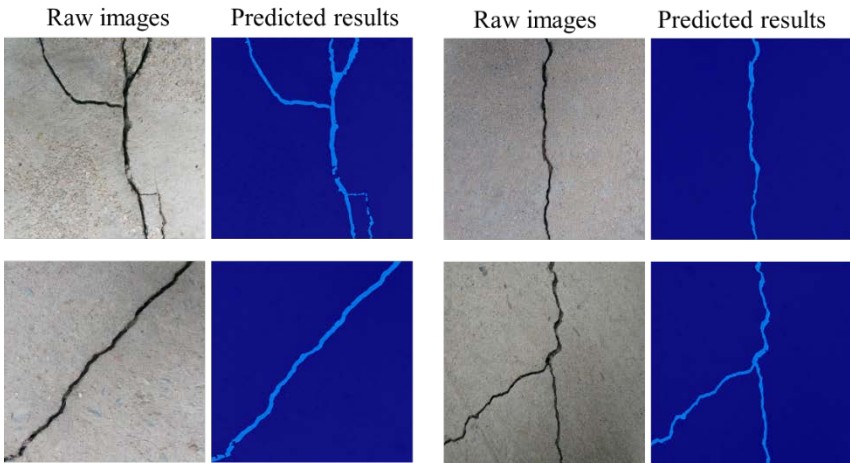

**Figure 11.** Detection results of the DeepLab_v3+ with 'resnet50'.

Table 4 shows the detection results of the SegNet, FCN, U-Net, and improved DeepLab_v3+. The MIoU and F-score of DeepLab_v3+ were at 0.84 and 0.93, nearly all higher than those of SegNet, FCN, and U-Net. The detection time for DeepLab_v3+ was at 1783 s, which was 25.2%, 46.4%, and 68.6%, respectively, lower than those of SegNet,

FCN and U-Net. These means that the detection accuracies of the SegNet, FCN, and U-Net are all lower than that of the improved DeepLab_v3+ despite the same accuracy (0.99) for four of them, and the detection time cost is higher than that of the improved DeepLab_v3+. Additionally, the detailed detection precision and detection speed ranking is in the same descending order as: improved DeepLab_v3+ > SegNet > FCN > U-Net. That is, a higher detection time does not mean a higher detection precision, and improved DeepLab_v3+ provides a fast and high precision network model for crack segmentation. The improved method proposed in this study is the best choice in considering the three precision indicators (MIoU, accuracy, and F-score) and calculation cost comprehensively, and this reversely proves the correctness of selecting resnet50 from the five backbone networks.

**Table 4.** Detection results of popular pixel-level segmentation networks.

| Evaluation Indicators | Popular Pixel-Level Segmentation Network | | | |
| --- | --- | --- | --- | --- |
| | **SegNet** | **FCN** | **U-Net** | **Improved DeepLab_v3+** |
| MIoU | 0.84 | 0.78 | 0.77 | 0.84 |
| Accuracy | 0.99 | 0.99 | 0.99 | 0.99 |
| F-score | 0.91 | 0.91 | 0.87 | 0.93 |
| Detection time (300 images) | 2383 s | 3326 s | 5679 s | 1783 s |

The DeepLab_v3+ network was subsequently used to detect the fixed and registration image, respectively. Finally, the change ratios of crack pixels were obtained. Part of the detection results are shown in Figure 12, and most of the errors (92% of the samples) were less than 0.3%. The detailed detection results are shown in Figure 13, and the average error is 0.11%. These results prove the feasibility of the approach proposed in this study based on image registration and the improved DeepLab_v3+.

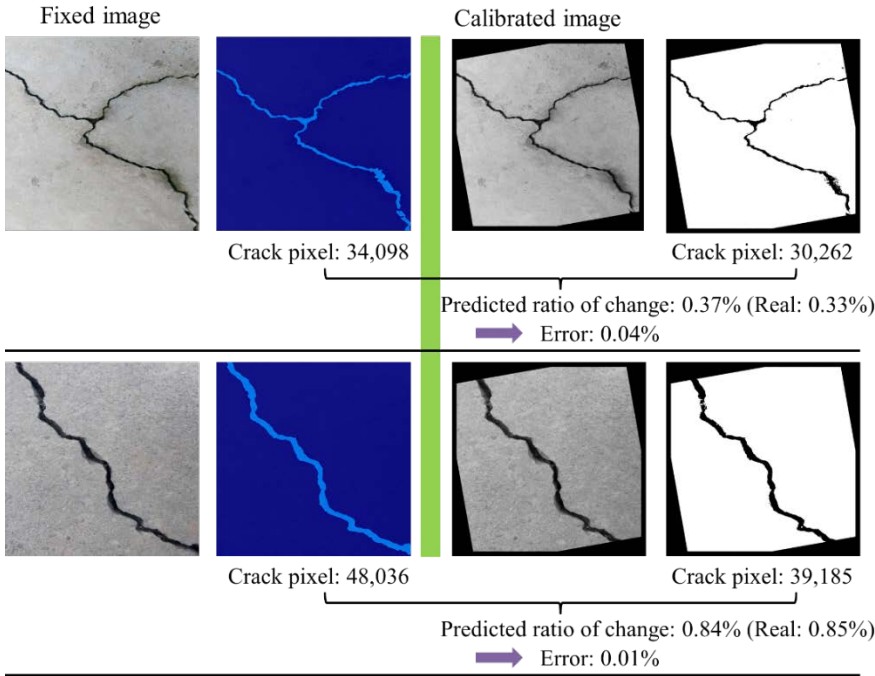

**Figure 12.** The change ratio of crack pixels.

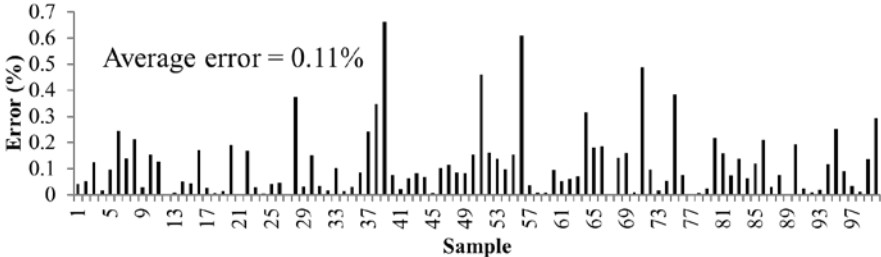

**Figure 13.** Errors of crack change ratio between the fixed image and calibrated images.

*3.3. Analyses of Parameters*

The neural network adopts gradient descent algorithms to obtain the optimal weights (w) of the network, which brings the more accurate prediction results. The most used gradient descent algorithms included stochastic gradient descent with momentum (sgdm), root mean square propagation (rmsprop), and adaptive moment estimation (adam), and the details were described in the reference [35]. In order to explore the most suitable gradient descent algorithm for crack detection, the sgdm, rmsprop, and adam were used to train the improved DeepLab_v3+, respectively. The testing results are shown in Table 5, and the training process is shown in Figure 14.

**Table 5.** Detection results of the improved DeepLab_v3+ using different algorithms.

| Evaluation Indicators | Gradient Descent Algorithms | | |
|---|---|---|---|
| | **Sgdm** | **Rmsprop** | **Adam** |
| MIoU | 0.84 | 0.73 | 0.57 |
| Accuracy | 99% | 98% | 95% |
| F-score | 0.93 | 0.81 | 0.60 |
| Training time (450 iterations) | 11,259 | 11,724 | 11,936 |

As shown in Table 5, the specific MIoU, accuracy, and F-score of sgdm were at 0.84, 99%, and 0.93, while those of rmsprop and adam were just at 0.73, 98% and 0.81, and 0.57, 95% and 0.60, respectively. That is, the MIoU, accuracy and F-score of sgdm were all higher than those of rmsprop and adam. This indicates that the descending order of detection precision is: Sgdm > rmsprop > adam. In addition, the training times for sgdm, rmsprop, and adam were at 11,259, 11,724, and 11.936, which were very close with each other. This indicates the slight influence of the optimizer on the training time. Therefore, the sgdm optimizer is the best choice of gradient descent algorithm.

The learning rate is another important parameter in deep learning because it determines whether and when the loss function will converge to the local minimum. In this study, the influences of four network models with different learning rates on the detection results were compared, and the learning rates were set as 0.1, 0.01, 0.001, and 0.0001, respectively. The network training process is shown in Figure 15, and the detection results are shown in Table 6. The results show that different learning rates lead to significant differences between MIoU and F-score, while the accuracy is very close. For example, the MIoU and F-score were at 0.72 and 0.85 with the learning rate at 0.1, while they increased to 0.84 and 0.93 when the learning rate was changed to 0.0001. The accuracy for the learning rate at 0.01, 0.001, and 0.0001 were all at 99% despite it being 98% for the learning rate at 0.01. Furthermore, the network model with the 0.001 learning rate has the highest MIoU, accuracy, and F-score, specifically at 0.84, 99%, and 0.93, which means this model has the highest detection precision. Additionally, the training time for the four set learning times were at 10,635 s, 10,439 s, 11,259 s, 11,235 s, respectively, suggesting that the learning rate does not affect the training time dramatically. Through the parametric analyses, the optimal optimizer and learning rate are determined, which can improve the efficiency and accuracy of the approach proposed in this study.

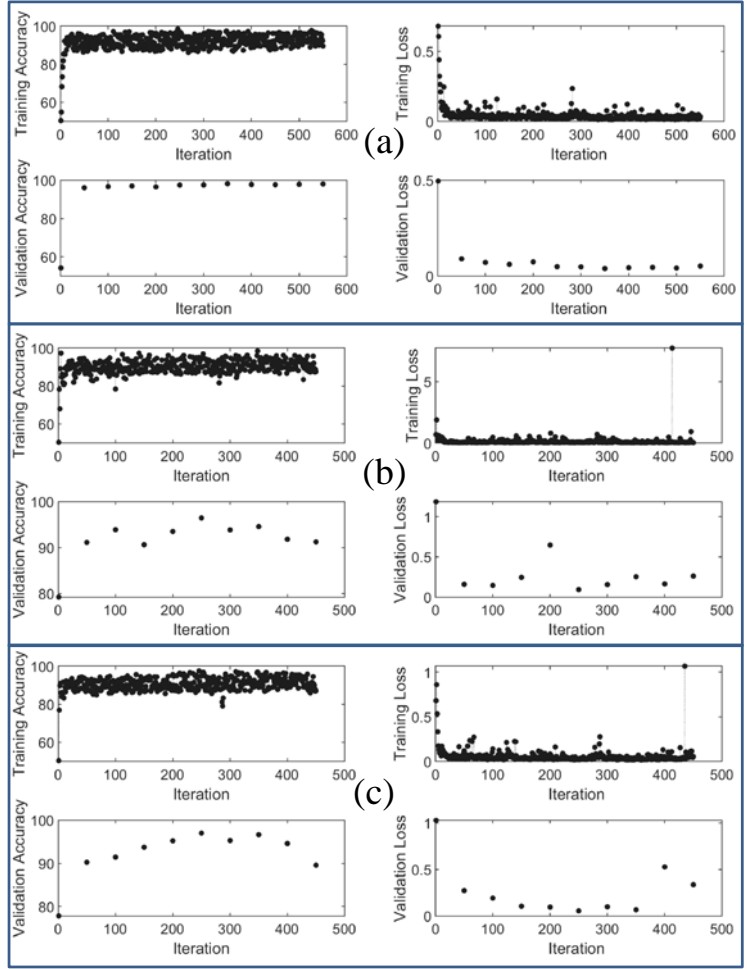

**Figure 14.** Training process of the DeepLab_v3+ using different gradient descent algorithms. (**a**) sgdm, (**b**) rmsprop, and (**c**) adam.

**Table 6.** Detection results of the DeepLab_v3+ using different learning rates.

| Evaluation Indicators | Learning Rate | | | |
|:---:|:---:|:---:|:---:|:---:|
| | **0.1** | **0.01** | **0.001** | **0.0001** |
| MIoU | 0.72 | 0.80 | 0.84 | 0.82 |
| Accuracy | 98% | 99% | 99% | 99% |
| F-score | 0.85 | 0.90 | 0.93 | 0.91 |
| Training time | 10,635 s | 10,439 s | 11,259 s | 11,235 s |

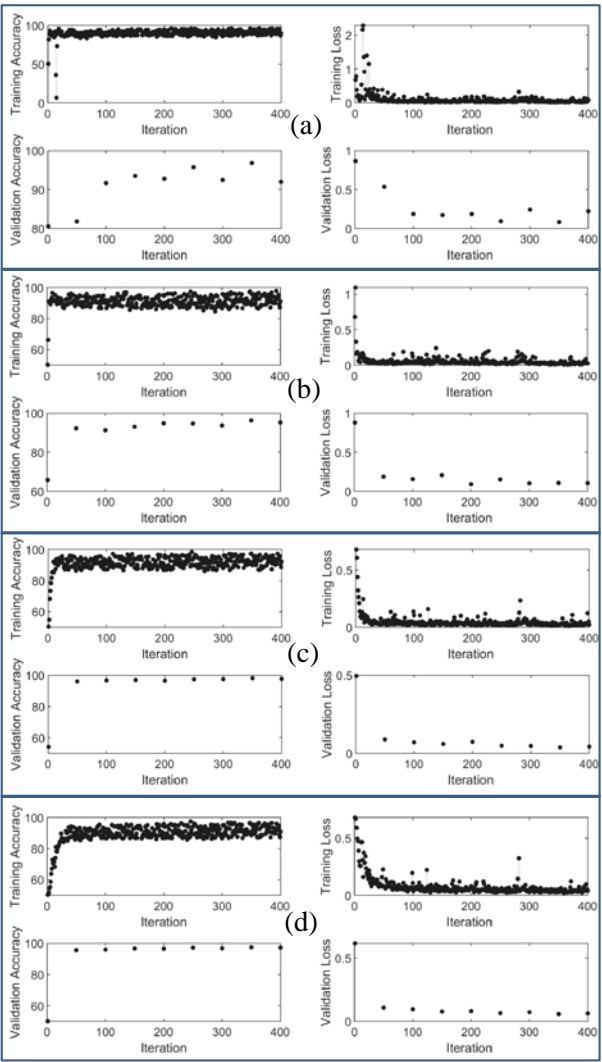

**Figure 15.** Training process of the DeepLab_v3+ using different learn rates. (**a**) 0.1, (**b**) 0.01, (**c**) 0.001, and (**d**) 0.0001.

## 4. Conclusions

In this study, a new detection approach of crack changes was proposed based on the combination of image registration and automatic crack detection (improved DeepLab_v3+). The cracks were detected and the change ratios were obtained after the image calibration. Compared with other popular detection algorithms and the latest approaches, the method proposed in this study showed great advantages in improving the precision of crack detection. In addition, the influence of the two optimizers and the learning rate of improved DeepLab_v3+ on the training results was also studied and the optimal optimizer and learning rate were confirmed. Based on the above research results, this study draws the following conclusions:

1.  Image registration technology can achieve the calibration of fixed and moving crack images accurately with a high similarity;
2.  Improved DeepLab_v3+ has a satisfying precision in pixel-level segmentation of cracks with the analyses of MIoU, accuracy, and F-score;
3.  The proposed crack change detection approach based on image registration and pixel-level segmentation performed well with a negligible average error of 0.11%;
4.  For the pixel-level segmentation of cracks, the most suitable optimizer and learning rate of the improved DeepLab_v3+ are sgdm and 0.001, respectively.

In practical engineering, it is often necessary to pay attention to the development of cracks (for example, the growth rate of cracks). The previous detection methods can only detect cracks and cannot obtain the development of cracks. The main challenge is that the cameras' positions of shooting cracks are often unfixed, so it is impossible to calculate the accurate change of cracks. The technical contribution of this paper is to realize the alignment of different spatial images through image registration technology, which is more suitable for the engineering needs. The application of this technology can effectively improve the detection level of building cracks and also provide an important support for the stable operation of building structures.

**Author Contributions:** Conceptualization, J.L. and Z.L.; methodology, Z.L. and S.T.; software, X.L. and S.T.; validation, X.L. and S.T.; formal analysis, Z.L.; investigation, Z.L.; resources, J.L.; data curation, Z.L.; writing—original draft preparation, S.T.; writing—review and editing, Z.L.; visualization, J.L.; supervision, Z.L.; project administration, Z.L.; funding acquisition, J.L. All authors have read and agreed to the published version of the manuscript.

**Funding:** This research received no external funding.

**Institutional Review Board Statement:** Not applicable.

**Informed Consent Statement:** Not applicable.

**Data Availability Statement:** Not applicable.

**Conflicts of Interest:** The authors declare no conflict of interest.

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
