# Peer review of "A New Approach to Automatically Calibrate and Detect Building Cracks"

_buildings, doi:10.3390/buildings12081081_

Round 1

Reviewer 1 Report

I appreciate the author for their good research work. 

The author just needs to discuss their results with their previous research.

Author Response

Revisions and Explanations

Manuscript Number:  buildings-1799038  

Title:  A new approach to automatically calibrate and detect building cracks   

Authors:  Zongchao Liu, Xiaoda Li, Junhui Li *, Shuai Teng *  

The authors are grateful for the reviewers' efforts in improving this paper. We have substantially revised the manuscript according to their valuable comments.

Responses to Reviewer 1

I appreciate the author for their good research work.

Response: The authors greatly appreciate the reviewer for the positive comments and valuable suggestions to improve the manuscript.

  • “The author just needs to discuss their results with their previous research.”

Response: Thank you for your comments. We have added comparisons about the proposed and previous methods.

Revision: Page15, Lines 374-382, “In practical engineering, it is often necessary to pay attention to the development of cracks (for example, the growth rate of cracks). The previous detection methods can only detect cracks, and cannot obtain the development of cracks. The main challenge is that the cameras’ positions of shooting cracks are often unfixed, so it is impossible to calculate the accurate change of cracks. The technical contribution of this paper is to realize the alignment of different spatial images through image registration technology, which is more suitable for the engineering needs. The application of this technology can effectively improve the detection level of building cracks and also provide an important support for the stable operation of building structures.”

Table A. The previous detection methods of crack detection

Cases

Publication Year

Detection Scenario

https://doi.org/10.1080/10298436.2018.1485917

2020

Classification of pavement cracks

https://doi.org/10.1111/mice.12363

2018

Classification of building cracks

https://doi.org/10.1061/(ASCE)CP.1943-5487.0000799

2018

Location of rail surface cracks

https://doi.org/10.1111/mice.12497

2020

Location of concrete cracks

https://doi.org/10.1016/j.autcon.2021.103788

2021

Pixel-level detection of pavement cracks

https://doi.org/10.1111/mice.12533

2020

Pixel-level detection of pavement cracks

Reviewer 2 Report

To increase readability and understanding, consider the following:

1. Introduction —  Present figures on the issue; and,

3. Conclusions — Open the applicability of the scientific step forward to the sector.

Author Response

Revisions and Explanations

Manuscript Number:  buildings-1799038  

Title:  A new approach to automatically calibrate and detect building cracks   

Authors:  Zongchao Liu, Xiaoda Li, Junhui Li *, Shuai Teng *  

The authors are grateful for the reviewers' efforts in improving this paper. We have substantially revised the manuscript according to their valuable comments.

Responses to Reviewer 2

  • “To increase readability and understanding, consider the following: 1. Introduction — Present figures on the issue; and, 3. Conclusions — Open the applicability of the scientific step forward to the sector.”

Response:  Thank you for your comments. We have added a figure to illustrate the issue. We have also added more statements to the conclusion for future applicability.

Revision: Page15, Lines 374-382, “In practical engineering, it is often necessary to pay attention to the development of cracks (for example, the growth rate of cracks). The previous detection methods can only detect cracks, and cannot obtain the development of cracks. The main challenge is that the cameras’ positions of shooting cracks are often unfixed, so it is impossible to calculate the accurate change of cracks. The technical contribution of this paper is to realize the alignment of different spatial images through image registration technology, which is more suitable for the engineering needs. The application of this technology can effectively improve the detection level of building cracks and also provide an important support for the stable operation of building structures.”

Reviewer 3 Report

1. Abstract: "....... for warning building damage in time." Please revise this, it seems not an appropriate term used.

2. The fundamental issue of building inspection and crack monitoring (and detection) is not covered. Please add some theoretical backing in performing building inspection and the primary/basic crack classification (at least structural or non-structural crack). In performing building inspections, visual inspections always take place as a start. This is the fundamental stage in building diagnosis. Where does this new approach sit in the whole process? 

3. Building crack as a defect is common in all types of buildings. In detecting the crack, does it suggest whether a number of cracks incident contributes to the structural integrity or not? or does it analyze every crack detected, as an individual building defect? 

4. The whole imaging process and analysis, does it "detect" or "monitor" the crack? There is a big difference as to detect and to monitor the crack. Please explain these 2 processes clearly. 

5. Where do all the images come from? before it is analyzed. The measurement of the crack itself, the direction (diagonal, crazing, horizontal, etc) that constitutes whether it is a structural or non-structural crack; how this is being considered in this new approach?

6. To some extent, crack detection is not a critical stage in analyzing building defects because cracks are a typical scenario in building structures. This is unavoidable since buildings are exposed to weathering effects, different temperatures, and various factors that contribute to the occurrence of building defects. What is more critical is the monitoring of the building crack (does it lead to structural?); so how does this is being addressed in this paper? 

7. The proposed approach is superb and the technicality is good and easy to follow.

8. In conclusion, please address the fundamental issue highlighted above, put this new approach in a bigger perspective, and how it contributes to the building defects analysis in general. 

Author Response

Revisions and Explanations

Manuscript Number:  buildings-1799038  

Title:  A new approach to automatically calibrate and detect building cracks   

Authors:  Zongchao Liu, Xiaoda Li, Junhui Li *, Shuai Teng *  

The authors are grateful for the reviewers' efforts in improving this paper. We have substantially revised the manuscript according to their valuable comments.

Responses to Reviewer 3

  • “Abstract: "....... for warning building damage in time." Please revise this, it seems not an appropriate term used.”

Response: We have revised it to an appropriate term.

Revision: Page 1, Line 9. “Timely crack detection plays an important role in building damage assessment”.

  • “The fundamental issue of building inspection and crack monitoring (and detection) is not covered. Please add some theoretical backing in performing building inspection and the primary/basic crack classification (at least structural or non-structural crack). In performing building inspections, visual inspections always take place as a start. This is the fundamental stage in building diagnosis. Where does this new approach sit in the whole process?”

Response: Thanks for your valuable comment. We have added the theoretical backing of structural cracks and non-structural cracks in the revised version.

Response: The visual inspection is the beginning of building inspection; however, the collected data should be processed and analysed to evaluate the structural bearing capacity. The method proposed in this paper will play a role in the stage of data processing and analysis, accurately detecting cracks and calculating the change information.

Revision: Page 1, Lines 29-39 “Cracks of different degrees commonly appear in building structures. The structural cracks are mainly caused by inadequate bearing capacity of the structures. It is the characteristic of the structural damage initiation or the symptom of insufficient structural strength, which is relatively dangerous, and the cracks must be further analysed to avoid the following disasters. The non-structural cracks including temperature cracks and shrinkage cracks always have little impact on the bearing capacity of the structure. Although these non-structural cracks do not reach the dangerous level of building collapse, these non-structural cracks can cause leakage, corrosion, concrete carbonation, etc., resulting in the reduction of the durability of building components, and even a serious potential threat to the safety and reliability of the structure.”

  • “Building crack as a defect is common in all types of buildings. In detecting the crack, does it suggest whether a number of cracks incident contributes to the structural integrity or not? or does it analyze every crack detected, as an individual building defect?”

Response: Thank you for your valuable comments. The analysis of a single crack cannot evaluate the overall performance of the structure, however, through the analyses of all cracks and summarizing the information, the overall performance of the structure can be evaluated.

  • “The whole imaging process and analysis, does it "detect" or "monitor" the crack? There is a big difference as to detect and to monitor the crack. Please explain these 2 processes clearly.”

Response: Thank you for your comments. The authors agree with the reviewer. “Detection” is the identification of events at a certain point in time; “Monitoring” is the continuous observation of events over a period of time. The method proposed in this paper focuses on the change at two moments of the crack. Therefore,“detect” is more suitable.

  • “Where do all the images come from? before it is analyzed. The measurement of the crack itself, the direction (diagonal, crazing, horizontal, etc) that constitutes whether it is a structural or non-structural crack; how this is being considered in this new approach?”

Response: Thank you for your comments. The crack image comes from a standard dataset (https://pan.baidu.com/s/1z-y3GhsWmbqzezD-eZSL-A, password: z493). The images used in this study are collected under the condition of in-service of the structure. In this paper, the change rate of non-structural cracks is mainly considered, and the image data of structural cracks will be considered in the future data collection process.

  • “To some extent, crack detection is not a critical stage in analyzing building defects because cracks are a typical scenario in building structures. This is unavoidable since buildings are exposed to weathering effects, different temperatures, and various factors that contribute to the occurrence of building defects. What is more critical is the monitoring of the building crack (does it lead to structural?); so how does this is being addressed in this paper?”

Response: Thank you for your comment. As the reviewer mentioned, this paper uses intelligent algorithm and image registration technology to detect the changes of cracks, and then obtains the crack growth speed. The performance of the structure can be evaluated according to the growth speed of cracks, as soon as the growth rate of cracks is too fast; it can be regarded as the structural performance has decreased. .

  • “The proposed approach is superb and the technicality is good and easy to follow.”

Response: Thank you for your encouraging comments.

  • “In conclusion, please address the fundamental issue highlighted above, put this new approach in a bigger perspective, and how it contributes to the building defects analysis in general.”

Response: Thank you for your comments. We have revised the paper according to the relevant comments.
